# Analysing the Relationship between Nutrition and the Microbial Composition of the Oral Biofilm—Insights from the Analysis of Individual Variability

**DOI:** 10.3390/antibiotics9080479

**Published:** 2020-08-04

**Authors:** Kirstin Vach, Ali Al-Ahmad, Annette Anderson, Johan Peter Woelber, Lamprini Karygianni, Annette Wittmer, Elmar Hellwig

**Affiliations:** 1Institute of Medical Biometry and Medical Statistics, Faculty of Medicine and Medical Center, University of Freiburg, Stefan-Meier-Str. 26, D-79104 Freiburg, Germany; 2Center for Dental Medicine, Department of Operative Dentistry and Periodontology, Faculty of Medicine and Medical Center, University of Freiburg, Hugstetter Straße 55, D-79106 Freiburg, Germany; ali.al-ahmad@uniklinik-freiburg.de (A.A.-A.); annette.anderson@uniklinik-freiburg.de (A.A.); johan.woelber@uniklinik-freiburg.de (J.P.W.); elmar.hellwig@uniklinik-freiburg.de (E.H.); 3Clinic for Conservative and Preventive Dentistry, Center of Dental Medicine, University of Zurich, Plattenstrasse 11, CH-8032 Zurich, Switzerland; Lamprini.Karygianni@zzm.uzh.ch; 4Institute of Medical Microbiology and Hygiene, Department of Medical Microbiology and Hygiene, Faculty of Medicine and Medical Center, University of Freiburg, Hermann-Herder-Straße 11, D-79104 Freiburg, Germany; annette.wittmer@uniklinik-freiburg.de

**Keywords:** microbiome analysis, bacterial concentration, oral biofilm, culture technique, inter-individual variation, sample size, study planning

## Abstract

The influence of a change in nutrition on the oral microbiota are discussed in literature, but usually only changes of population mean values are reported. This paper introduces simple methods to also analyse and report the variability of patients’ reactions considering data from the culture analysis of oral biofilm. The framework was illustrated by an experimental study exposing eleven participants to different nutrition schemes in five consecutive phases. Substantial inter-individual variations in the individual reactions were observed. A new coherence index made it possible to identify 14 instances where the direction of individual changes tended to coincide with the direction of the mean change with more than 95% probability. The heterogeneity in variability across different bacteria species was limited. This allowed us to develop recommendations for sample sizes in future studies. For studies measuring the concentration change of bacteria as a reaction to nutrition change, the use of replications and analysis of the variability is recommended. In order to detect moderate effects of a change in nutrition on the concentration of single bacterial taxa, 30 participants with three repetitions are often adequate. Insights into the relationship between nutrition and the microbial composition can be helpful for the development of dietary habits that promote the establishment of a healthy microbial flora and can therefore prevent the initiation of oral diseases such as caries and periodontitis.

## 1. Introduction

In recent years, the influence of nutrition on oral microbiota was discussed in literature. While some papers investigate the nutrition-induced change of diverse periodontal parameters [1,2,3,4,5,6], others discuss the influence of nutrition on the oral microbiota [7,8,9,10,11,12]. Usually, the authors focus on the mean values of the individual changes and do not investigate their variation. However, even if there is a distinct mean change, this does not imply that all subjects show a similar change, and some subjects may even show a change in the opposite direction than indicated by the mean change. Information on the uniformity of the change in the background population of interest can assist in the interpretation of study results: the more uniform a change, the more we can regard an observed change in mean values as a “typical” change, which applies to the (vast) majority of a population. Analysing the individual variability of changes is the key to getting a better understanding of the degree of uniformity.

Analysing the observed variation of the individual changes is, however, misleading, as this variability also represents the measurement error or biological short term variation. The variability of the true, underlying changes, can be analysed by random effect models if repeated measurements are performed. Consequently, we present a framework to analyse the individual variability in the response to nutrition based on fitting random effect models and on reporting the estimated variability in a user-friendly manner. In particular, we suggest explicitly reporting an estimate of the fraction of subjects for whom the signs of the true change equal the sign of the mean change. Using the data from an experimental study, it is illustrated how these additional analyses can supplement a traditional analysis focusing on the mean change and its precision. In particular we observe that (statistically significant) mean changes in the magnitude of a half log step often still allow that 10% or more of the population will experience a change in the opposite direction.

Since the inter-individual variability of changes is also a major source of the overall variability determining the power and sample size of such studies, we also illustrate how an analysis of inter-individual variability may inform the planning of future studies. It turns out that the inter-individual variability of changes is rather similar across different bacteria, allowing us to develop rather general rules. Based on the analysis of the present study, it is recommended to include repeated measurements in any case, and that sample sizes in the magnitude of 30 participants and three repetitions are often adequate to detect moderate differences.

Keeping in mind that shifts in biofilm composition towards a dysbiotic microbiota could trigger the onset of oral diseases, one could assume that this study gives new statistical insights in the relationship between nutrition and the microbial composition. This can be helpful for the development of dietary habits that promote the establishment of a healthy microbial flora and can therefore prevent the initiation of oral diseases such as caries and periodontitis. The report can also be indirectly helpful for the development of antibacterial compounds and alternative treatment methods such as antimicrobial photodynamic therapy (aPDT) targeted against specific oral bacteria or oral biofilm. Alternative biofilm treatment methods such as aPDT or natural compounds may alter the oral biofilm composition as well.

### 1.1. Materials and Methods

In this study, 11 adults (5 male and 6 female) aged between 21 and 56 years, the data concerning nutrition and the microbiota in their oral biofilm were collected over 15 months [13]. The participants ran through 5 phases, each of which lasted 3 months, following a specific nutritional protocol. In a first lead-in-phase, the participants kept their regular diet, which served as baseline. Subsequently, the participants changed to a three-month-long diet (phase 2) with an additional daily consumption of 10 g sucrose in the form of small pieces of rock candy (2 g) 5 times between meals. In phase 3, the nutrition was changed to meals containing milk protein. The yoghurt and milk that were used contained both 1.5% fat and were purchased from Schwarzwaldmilch GmbH, Freiburg, Germany. In addition to their normal food, they ate 150 g of yoghurt three times a day and drank 100 mL milk twice a day. Both the yoghurt and the milk were evenly distributed in the oral cavity and left there for an exposure time of three minutes.

In phase 4, the nutrition was changed to a high-fibre diet: the participants consumed a total of 500 g of vegetable puree, which again was evenly distributed in the oral cavity and left there for an exposure time of three minutes. In the final phase (phase 5), the participants returned to their normal diet like in the lead-in-phase.

An in-situ splint system was used to sample the dental plaque. Individual upper jaw rigid acrylic appliances were manufactured for each study participant [13]. The splint system was worn at all times except during meals and dental hygiene. During all phases, the dental biofilm was allowed to grow on embedded enamel slabs over the course of seven days. Subsequently, the splint system was removed for analysis of the dental biofilm, cleaned, and after seven days it was re-applied for another seven days. This procedure was repeated three times, resulting in three measurement points per phase (Figure 1 based on an earlier version in [13]).

The previous study [13] refers to the same patient collective, but used a different technique (sequencing) to measure the bacterial composition, thereby including only two diet phases in the analysis. The study showed that in the sucrose phase (phase 2) the microbial community composition was significantly different than in phase 1 (baseline). Especially the abundance of oral streptococci was significantly increased.

For our study, the bacteria were isolated and identified as previously described in detail [14,15]. The vials containing the samples were thawed at 36 ${}^{\circ }C$ in a water bath and vortexed for 30–45 s. For the isolation and identification of the microorganisms, 100 μL of the undiluted sample and serial dilutions thereof were plated on different agar plates. The serial dilutions ($10^{-1}$ to $10^{-7}$) were prepared in peptone yeast medium (PY). Each dilution was plated on Columbia blood agar plates (CBA) at 37 ${}^{\circ }C$ and 5%–10% CO_2_ atmosphere for 5 days to cultivate aerobic species and on yeast-cysteine blood agar plates (HCB) to cultivate anaerobic bacteria at $37^{\;\circ }C$ for 10 days. The grown bacterial colonies were phenotypically evaluated and counted. Subsequently the number of colony-forming units (CFUs) per ml in the original sample was calculated. Pure cultures of all colony types were sub-cultivated and analysed by MALDI-TOF (MALDI Biotyper, Bruker Daltonik GmbH, Bremen, Germany), as described earlier in detail [16]. The dental samples reached concentration values between 0 and $4.72\times 10^{9}$ CFU/mL. For the analysis the data were log-10-transformed. The values were set to 4 if the concentration values were below the detection limit of 4. We considered the following microorganism groups (further details and abbreviations in the Appendix B): aerobic bacteria (faecal contaminants were excluded), aerobic bacteria with faecal contaminants, faecal contaminants, anaerobic bacteria, all with faecal contaminants, all (faecal contaminants were excluded), *Streptococcus oralis* 1, *Streptococcus oralis* 2, *Streptococcus oralis* 3, *Streptococcus mutans*, *Gemella Granulicatella Streptococcus* species pluralis (spp.), *Actinomyces* spp., *Rothia* spp., *Lactobacillus vaginalis, Neisseria* spp., *Capnocytophaga* spp., *HACEK, fungi*, black pigmented Bacteroides spp., non-pigmented Bacteroides spp., *Fusobacterium* spp., *Campylobacter* spp., *Selenomonas* spp., Gram-positive aerobic cocci, Gram-positive aerobic rods, Gram-negative aerobic cocci, Gram-negative aerobic rods, Gram-positive anaerobic cocci, Gram-positive anaerobic rods, Gram-negative anaerobic cocci and Gram-negative anaerobic rods. For some of them (*Streptococcus mutans*, *Lactobacillus vaginalis, fungi*, black and not pigmented bacteria, *Selenomonas* spp., Gram-positive anaerobic cocci and rods), the percentage of values below the detection limit was greater than 75%. We decided to exclude these bacteria due to their limited variation. In the following paragraphs we refer to each of these groups as one bacterial category.

#### 1.1.1. Inter-Individual Variation and Modelling

In this kind of study, two types of variability are of interest for each bacterial taxa: the inter-individual variability or “normal population variation” of the concentration in the individual diet and the variability of the individual reaction to a nutrition change. The “normal population variation” reflects how much variation we can expect due to inter-individual variations in eating habits, oral microbiota, etc. However, our main interest lies in how uniformly the participants react to a change in diet. The more uniform the reaction, the better we can generalise a change observed in mean values to all patients. A naive way to analyse the “normal” population variation is to consider the empirical variation across individuals in our phase 1, where there was no change in diet. Similarly, we can analyse the empirical variation of the changes in bacterial concentration after the change in diet. However, both empirical variations also reflect measurement errors and biological short term variation. Fortunately, in our study, there are built-in repetitions within each phase, which enables a direct estimate of the inter-individual variation by appropriate modelling.

In the following, we denote the eleven participants with *i*, the five phases with *p* and the three repetitions with *r*. We consider the following model for the log-10-transformed concentrations $y_{ipr}$ of a bacterial category
(1)$$y_{ipr}=\alpha_{i}+\Delta_{ip}+\epsilon_{ipr}$$
with $\alpha_{i}$ reflecting the individual initial level of phase 1, $\Delta_{ip}$ the incremental changes compared with phase 1 and $\epsilon_{ipr}$ representing the measurement error and individual biological variability. We further assume $\alpha_{i}∼N\left(\mu_{\alpha },\sigma_{\alpha }^{2}\right)$, $\Delta_{ip}∼N\left(\mu_{\Delta_{p}},\sigma_{\Delta_{p}}^{2}\right)$, $\epsilon_{ipr}∼N\left(0,\sigma_{res}^{2}\right)$ for p=2,…,5 and independence of these quantities. The parameters of interest are the initial mean value $\mu_{\alpha }$, the mean values of the incremental changes $\mu_{\Delta_{p}}$ per phase, the standard deviation $\sigma_{\alpha }$ describing the inter-individual variability at the initial level and the standard deviations $\sigma_{\Delta_{p}}$ characterising the variability in the individual reaction to a change in diet for each phase. Given that the phase-specific standard deviations $\sigma_{\Delta_{p}}$ are difficult to estimate due to the small sample size, a model based on the assumption that all standard deviations $\sigma_{\Delta_{p}}$ are equal to a common value $\sigma_{\Delta }$ is also fitted. For model fitting, we use the REML technique [17].

#### 1.1.2. Guidance to Interpretation

Standard deviations are often difficult to interpret for non-statisticians. Therefore, we make use of three techniques to assist non-statisticians in their interpretation. First, if a random variable *z* is drawn from a normal distribution with mean μ and a standard deviation σ, the interval μ±1.96σ covers 95% of all observations of *z*. We refer to this as the 95% range of *z*. We can apply this to both the initial values ($z=\alpha_{i},\mu =\mu_{\alpha },\sigma =\sigma_{\alpha }$) as well as the increments ($z=\Delta_{i},\mu =\mu_{\Delta_{p}},\sigma =\sigma_{\Delta }$). Secondly, we consider the probability that a single observation of *z* is above a constant *c*, which is given by
(2)$$P\left(z>c\right)=1-\Phi \left(\frac{c-\mu }{\sigma }\right).$$

We apply this to the incremental changes with c=0, as in this instance we are interested in knowing the probability of a positive (or negative) reaction to a change. More precisely we are interested in a measure that reflects the coherence of individual changes with the mean change. Accordingly, if μ>0 we are interested in P(z>0), if  μ<0 in P(z<0).

Consequently, we define the coherence $\eta_{p}$ of the individual changes with the mean change for each phase *p* as:(3)$$\eta_{p}=\left\{\begin{array}{cc} 1-\Phi (\frac{-\mu_{\Delta_{p}}}{\sigma_{\Delta_{p}}}) & \mathrm{if}\;\mu_{\Delta_{p}}>0\\ \Phi (\frac{-\mu_{\Delta_{p}}}{\sigma_{\Delta_{p}}}) & \;\mathrm{otherwise}. \end{array}\right.$$

Thirdly, we consider the expected absolute difference for two randomly-chosen values $z_{1}$ and $z_{2}$, which is equal to 1.13 times the standard deviation (a derivation is given in the Appendix C). We denote this in the following with $E_{diff}$. For example, if we apply this to the true initial values $\alpha_{i}$, $E_{diff}$ is equal to the difference that can be expected if we consider two randomly chosen individuals.

#### 1.1.3. Heterogeneity in Individual Variation

To use results on inter-individual variability with respect to the planning of studies, it is desirable to derive conclusions that are valid independent of the choice of the bacterial category. Hence, it is of interest to investigate the heterogeneity of variations across the bacteria. We approach this by random effects meta-analyses of the estimated standard deviations considering each bacterial category as a “study”.

Such a meta-analysis is based on the model
(4)$$log\;\;{\hat{SD}}_{b}=log\;\;SD_{b}+\epsilon_{b}\;\;\;\mathrm{with}\;\;\;\epsilon_{b}∼N\left(0,\sigma_{b}^{2}\right)\;\;\;\mathrm{and}\;\;\;log\;\;SD_{b}∼N\left(\mu ,\tau^{2}\right)$$
where $log\;\;SD_{b}$ is the logarithm of the true standard deviation of a bacterial category *b*, $\sigma_{b}$ the true standard error of $log\;\;{\hat{SD}}_{b}$, which we replace by its estimate, μ the average logSD and τ the standard deviation of the true logSD values. τ allows us to judge the heterogeneity of the true standard deviations between the bacteria. In particular, with the technique explained above we can build a 95% range for the true standard deviations based on τ that illustrates the variation. In addition, we will try to identify causes for the variation, including an examination of the relationship to the initial mean value.

#### 1.1.4. Sample Size

We will use the results of our study to perform a sample size calculation for further studies. If we are interested in a sample size calculation for a study examining the results of two different diets on the bacterial concentration of a single strain of bacteria in a paired design, we can link this scenario to our study comparing phase p with the initial phase. The mean value $\mu =E[\theta_{i}]$ and the variance $\sigma^{2}=Var\left(\theta_{i}\right)$ of the individual differences $\theta_{i}={\bar{y}}_{ip}-{\bar{y}}_{i1}$ have to be specified for a sample size calculation.

In the case of *R* repetitions per phase and participant, we have
(5)$$\begin{array}{c} \begin{array}{cc} \sigma^{2} & =Var({\bar{y}}_{ip}-{\bar{y}}_{i1})\\ \;\; & =Var\frac{1}{R}\sum_{r=1}^{R}\left(\alpha_{i_{1}}+\Delta_{ip}+\epsilon_{ipr}-\alpha_{i_{1}}-\epsilon_{i1r}\right)\\ \;\; & =\frac{1}{R^{2}}Var\left(R\Delta_{ip}+\sum_{r=1}^{R}\epsilon_{ipr}-\epsilon_{i1r}\right)\\ \;\; & =\frac{1}{R^{2}}\left(R^{2}\sigma_{\Delta_{p}}^{2}+2R\sigma_{res}^{2}\right)\\ \;\; & =\sigma_{\Delta_{p}}^{2}+\frac{2}{R}\sigma_{res}^{2}\\ \; \end{array} \end{array}$$

The choice of μ is discussed later. The sample size for a power of 0.9 and a significance level of α=0.05 is computed using the formula by Chow, Shao and Wang [18].

#### 1.1.5. Software

For the analyses, the statistics program STATA (StataCorp LT, College Station, TX, USA, Version 15.1) was used. For estimates of means and standard deviations, the  xtmixed procedure with the option reml was applied after rewriting model (1) in terms of fixed and random effects. Meta-analyses were performed with the method by DerSimonian and Laird [19] using a random effects model, provided in STATA as the metan command with the options random and eform for log-transformed values. For graphical presentation, bar charts, scatter plots and forest plots were used.

## 2. Results

### 2.1. Traditional Analysis

Figure 2 shows the mean changes of log bacteria concentrations in comparison to baseline with 95% CI intervals for each of the four phases. Within each of the phases 2 to 4, we observe that the majority of bacteria show no or a rather small mean change of less than 0.3 log steps, whereas a few bacteria always show changes in the magnitude of a half log step or above. For phase 2, i.e., the sucrose phase we observe a mean increase of more than 0.5 log steps for *Rothia* spp. which may hint at a possible role of this genus in early caries development. The results of [13] could not be confirmed.

For phase 3, i.e., the dairy phase, we observe a mean decrease of more than 0.5 log steps for *Neisseria* spp., *Capnocytophaga* spp., HACEK, Faecal contaminants and Gram-negative aerobic cocci and rods, which may hint at the decreased ability of these bacterial taxa to metabolize the nutrients that are available in this phase. Additionally, a mean increase of more than 0.5 log steps for *Actinomyces* and *Rothia* spp. was observed.

In the dietary fibre phase (phase 4), a mean decrease of more than 0.5 log steps for *Neisseria* spp., *Capnocytophaga* spp., HACEK, Faecal contaminants and Gram-negative aerobic cocci and rods and a mean increase of more than 0.5 log steps for *Capnocytophaga* spp. were detected.

In phase 5, returning to the regular diet of the participants, the most pronounced pattern of mean changes was observed, with mean decreases of more than one log step for *Neisseria* spp., *Capnocytophaga* spp. and Gram-negative aerobic cocci and rods. Altogether, different fluctuations were observed for all phases reflecting the high dynamic of the oral microbiota within the supragingival oral biofilm.

The distinct patterns observed in Figure 2 for each phase may suggest that this distribution might be typical for each study participant. To which degree this is justified is investigated in the following.

### 2.2. Illustrative Applications

First, we illustrate the application of our approach based on the example of two specific bacterial categories: anaerobic bacteria and *Rothia* spp. The raw data actually analysed are shown in Figure 3. Table 1 presents the results.

For anaerobic bacteria, we obtain an initial mean value ${\hat{\mu }}_{\alpha }$ of 6.64 and a ${\hat{\sigma }}_{\alpha }$ of 1 describing the inter-individual variability of the initial level. This means the individual initial values have a 95% range of [4.68,8.60]. Since we used logarithmic values, this implies that the participants at opposing ends of the distribution have a difference of up to four log steps.

The ${\hat{\mu }}_{\alpha }$ as well as the ${\hat{\sigma }}_{\alpha }$ of *Rothia* spp. is lower, with values at 5.83 and 0.44 respectively; hence, the initial values are much more homogenous, which can also be seen in the smaller 95% range of [4.97,6.69]. Next, we consider how uniformly the participants react to a change in diet. For anaerobic bacteria, the largest mean change can be observed from phase 1 to phase 5 with −0.741, while for *Rothia* spp. the largest change occurs between phase 1 and phase 2 with a value of 0.639. The variability in the individual reaction to a change was first estimated for each of the phases ( ${\hat{\sigma }}_{\Delta_{2}},{\hat{\sigma }}_{\Delta_{3}},{\hat{\sigma }}_{\Delta_{4}},{\hat{\sigma }}_{\Delta_{5}}$). These estimates are quite unstable, as can be seen in the wide confidence intervals. Therefore, we prefer to consider the common standard deviation for all increments ${\hat{\sigma }}_{\Delta }$, which we can estimate with higher precision and thus smaller confidence intervals. For anaerobic bacteria, we observe a ${\hat{\sigma }}_{\Delta }$ of 0.54 and for *Rothia* spp., a smaller individual change of 0.29, which suggests that the effects for *Rothia* spp. are more homogenous. We can combine this with the mean values observed at different phases. For anaerobic bacteria, we obtain a 95% range for the individual increments from phase 1 to phase 5 of [−1.80, 0.32] with 0 inside the range, while for *Rothia* spp. the 95% range for the individual increments from phase 1 to phase 2 is narrower with [0.07, 1.21].

If we look at the coherence $\eta_{p}$ of the individual increments with the mean effect for each phase, for anaerobic bacteria we observe only one value above 90%, while for *Rothia* spp. for all phases with the exception of phase 4 high values are reached. This can also be seen in the plots (Appendix A) for *Rothia* spp. nearly all participants show the same reaction to a change in diet in each phase in the form of a decrease or increase, while for anaerobic bacteria different reactions are observed.

### 2.3. Systematic Application

Table 2 and Table 3 present the results for all bacteria classes. Figure 4a depicts the estimated standard deviations for the increments $\sigma_{\Delta }$, which are always below 1. There are some differences between the bacteria, but most standard deviations are around 0.5. The lowest standard deviations are observed for faecal contaminants, *Gemella Granulicatella Str.* spp. and *Rothia* spp., indicating that these bacteria change to a more similar degree for all participants. There seems to be some association between the standard deviation of the baseline values and the standard deviation of the increments (Figure 4b), suggesting that bacteria more variable in their concentration in the population also tend to be more variable in their increments across the participants (correlation coefficient r=0.56). In contrast, there is no association between the standard deviations and the mean values (Appendix A), i.e., bacteria appearing at higher or lower concentrations do not tend to be more or less variable, respectively.

In Table 3 we observe that η values are often close to 0.5, i.e., the lowest possible value. However, this is not surprising for bacteria with a mean change close to 0, as then it is obviously a chance result whether a single participant follows the mean trend or not. It is, however, of interest to focus on those bacteria for which we could observe in Figure 2a distinct mean increase or decrease in some phases. These bacteria and phases are shaded gray in Table 3. In 14 out of 20 instances, the η values are above 95%, indicating that in future participants, we can expect that the vast majority has a change in the concentration as indicated by the change in mean values. However, there are some exceptions. For *Capn* spp. in phase 3 and C. spp. in phase 4, we observe only values of 85.8% and 82.8%, respectively, indicating that roughly each 6th participant will show a change in concentration opposite to the change indicated by the mean values. In three other instances, we reach values close to 90%, indicating that this happens in every 10th participant. A similar picture can be observed when focusing on instances where the deviation in mean change was significantly different from 0 in the standard analysis (marked with a * in Table 3.) The minimal η value observed here is 79.9%.

In summary, these analyses show that we should not interpret a distinct or significant mean change as an indicator that a change in this direction will happen for all subjects in the background population if exposed to a corresponding nutrition. It is possible to quantify the extent to which this holds true by η values, and hence they may add information to an analysis.

### 2.4. Heterogeneity

#### 2.4.1. Heterogeneity in the Standard Deviation of the Initial Values

A meta-analysis of the estimated standard deviation ${\hat{\sigma }}_{\alpha }$ of the initial values (Figure 5) was performed. Each bacterial category here corresponds to a study. Four studies in which the standard error was not estimable were excluded. The most important outcome of this meta-analysis is that we obtain an estimated τ of 0, which means that there is no evidence of heterogeneity across the different bacterial groups. As the result of the meta-analysis, we obtain an overall value for $\sigma_{\alpha }$ of 0.79, which means that the initial values have a typical range of three log steps (mean initial value ±0.79×1.96).

#### 2.4.2. Heterogeneity in the Standard Deviation of The Increments

Looking at the meta-analysis of the standard deviations of the increments (Figure 6), we again observe a τ of 0. Larger confidence intervals can be found for the bacteria 3, 10 and 12. The overall value of $\sigma_{\Delta }$ is 0.5, which we used for sample size calculations.

#### 2.4.3. Heterogeneity in the Standard Deviation of The Residuals

In the meta-analysis of the standard deviations of the residuals (Figure 7), we obtain a τ of 0.26. With the exception of numbers 1, 2, 5, 6, 7 and 18, we observe rather homogenous values. Looking at Figure 8, we observe smaller residuals for bacterial groups with a large initial mean value. These bacterial groups correspond to the aforementioned bacterial groups. This is not particularly surprising because they do not represent single bacteria but rather relatively large bacterial groups. Hence, the residuals are smaller due to averaging over many bacteria. We obtain an overall $\sigma_{res}$ of 0.87 with a 95% range of [0.52, 1.45] for $\sigma_{res}$.

If we exclude the bacterial groups with large initial mean values, we observe a smaller τ of 0.11 and an overall $\sigma_{res}$ of 1.0 with a 95% range of [0.79, 1.21] for $\sigma_{res}$. For the bacteria groups with large initial mean values, we obtain a τ of 0.14 and an overall $\sigma_{res}$ of 0.59 (95% range [0.32, 0.86]). To illustrate the influence of $\sigma_{res}$ on the sample size calculation, both values 0.6 (for larger bacterial groups) and 1.0 (for single bacteria) were used.

### 2.5. Sample Size Considerations

As pointed out in the Materials and Methods, for a study comparing two diets in a paired design, the variance of the final estimates depends on the population standard deviation $\sigma_{\Delta }$, the standard deviation of the residuals $\sigma_{res}$ and the number of repetitions R. Equation (Equation 5) indicates that the variance decreases with increasing R, but in any case there remains the contribution of the population standard deviation. For a sample size consideration we have to specify values for the standard deviations $\sigma_{\Delta }$ and $\sigma_{res}$, for the mean differences μ and the repetitions *R*. Our considerations about the heterogeneity of the different standard deviations across the bacterial groups suggest using a $\sigma_{\Delta }$ of 0.5 and a $\sigma_{res}$ of 0.6 and 1, respectively. With respect to the choice of μ, we should take into account the clinical context. However, little is known about the impact of changes in the microbiota on the individual. From a statistical perspective, we argue that a relevant effect should explain some of the overall variation between individuals. A useful benchmark may be the expected difference in initial values between two randomly-chosen individuals. This is a simple function of the population standard deviation $\sigma_{\alpha }$, (namely 1.13 * $\sigma_{\alpha }$, as pointed out in the Appendix B). We have seen that there is no substantial variation in $\sigma_{\alpha }$, and hence we can use the estimated value ${\hat{\sigma }}_{\alpha }=0.79$, resulting in μ=0.89≈0.9. However, this value is rather large compared with effects observed in other studies [4,20] and what we observed as effect estimates in our study. In former studies, Tenuta et al. [20] found a change of biofilm in glucose + fructose and sucrose groups in comparison with a negative control group and Filoche et al. [3] found that plaque from different donors showed a different reaction to sucrose. The results of Tenuta for the mean concentration of *S. mutans* (log 10-transformed to be comparable with our results: negative control: 2.04; treatment with glucose + fructose: 2.81; treatment with sucrose: 2.63) indicate that changes in the range of a half log-10 step seem realistic. Therefore, we also considered the values 0.7 and 0.5.

In Table 4, we report the sample size for different combinations of μ, $\sigma_{\Delta }$, $\sigma_{res}$ and a varying number of repetitions *R*. The use of repetitions allows for a substantial decrease in the number of study participants: the use of two repetitions already leads to a reduction of approximately 40% and a slight increase in the number of measurements. If we look back at the variance formula in the methods part, we see that $\sigma_{res}\approx 2*\sigma_{\Delta }$ (thus $\sigma_{res}^{2}\approx 4*\sigma_{\Delta }^{2}$ ), resulting in $\sigma_{\Delta }$ having only a minor influence on the sample size. If costs per measurement are lower than the costs per study participant, even more repetitions can save costs. The overall sample size depends on the choice of μ. Our study ($R=3,\sigma_{\Delta }=0.5,\sigma_{res}=1$) was powered to detect a difference in the magnitude of 0.9. In order to also detect moderate differences, one should conduct more than 90 measurements.

## 3. Discussion

Our results show that an analysis of participants’ variation regarding bacterial concentration as a result of a change in diet is feasible and useful. The estimation of inter-individual variation is possible if some repetitions of the observations are made and allows for more information in analysing single bacteria.

In most studies, the authors usually look at the changes of the mean values and interpret them as general differences. Standard deviations of the incremental changes offer more insights. In particular, the coherence of the individual increments with the mean change allows for a better understanding of results. If we look at the $\eta_{p}$ for the single phases in Table 3, in two cases we even obtain values of $\eta_{p}<86\%$ for significant effects. On the other side there are also non-significant mean changes with a high coherence. In Table 5, we computed η for realistic values of μ and σ. Tables like this can help to classify one’s own results. We can see that with a small μ and a relatively large σ, no uniform reaction of subjects from the background population can be expected.

For the initial values, we typically observe an inter-individual variability corresponding to a 95% range of three log-10 steps. This is not particularly surprising because Aas et al. [7] have already observed differences of the bacterial flora in the healthy oral cavity, even between different oral sites of the same person.

The variation of increments was distinctly smaller, suggesting a 95% range in the magnitude of two log steps. Residuals showed a variation similar to the initial values. It is important to note that all these insights about variation and coherence are only possible, if the study design includes repeated measurements.

Nutrition can influence the oral biofilm composition and thus, the onset of oral diseases. Specifically, the content of fermentable carbohydrates is crucial in the process of cariogenic demineralisation [21]. However, recent studies also showed an influence of sugar on gingival inflammation which might be etiologically related to both local and systemic effects like elevated blood sugar [22,23]. Secondly, nutrition can have antibacterial and biofilm-inhibiting properties, which can be attributed to polyphenols [24]. Furthermore, nitrates can have both anti-cariogenic and anti-inflammatory effects on gingivitis [25,26].

Due to the homogeneously-estimated standard deviations of both the increments, the residuals and the initial values for different bacteria, the results can be used for planning new studies. If the concentration of bacteria as a reaction to a change in diet similar to the setup in our study should be examined, and a mean difference of half a log-10 step should be shown, we recommend recruiting 31 participants and three repetitions per phase. For the analysis of the data, the proposed 95% ranges and coherence measures η should be computed to make the results more clear. For some bacterial groups, we observed lower standard deviations of the residuals than for others due to the fact that we sometimes grouped bacteria together into sub-categories. This point should be incorporated into the planning of a study because smaller sample sizes are required for bacterial groups than for single bacteria.

Due to the small sample size, our study has some limitations. We have already argued above that the standard deviations of the increments for the single phases are quite unstable. Another limitation is that often values were below the detection limit. Due to the small sample size, the use of multilevel mixed-effects tobit regression for continuous responses, where the outcome variable is censored with the detection limit as a censoring limit was not possible. In this paper we focused on the analysis of single bacteria groups. However, understanding the change of the whole bacteria spectrum is also of interest. This requires extending the ideas presented in this paper to a multivariate setting. Our proposal supplements the recommendation for heterogeneity measures for logistic regression models by Larsen et al. [27]. Of course, as with any experimental study, we have to make sure that the participants behave according to the rules, on the one hand with regard to nutritional requirements and on the other hand with regard to wearing the splint. It has been shown in many previous own studies that supragingival oral biofilm was cultivated sufficiently on enamel slabs, which are fixed in splint systems. However, splints still remain a model system which simulates the natural conditions of biofilm formation in the oral cavity. It cannot be excluded that the salivary pellicle and the subsequent formation on bovine enamel slabs could be slightly altered. Additionally, during meals, the splint systems had to be taken out and stored in saline solution (0.9%).

## 4. Conclusions

For studies measuring the concentration change of bacteria as a reaction to nutrition change, the use of replications and analysis of the variability is recommended. Our suggestions contribute to a better understanding and reporting of the individual variation in bacterial concentrations and to a more targeted planning of new studies.

## Figures and Tables

**Figure 1 antibiotics-09-00479-f001:**
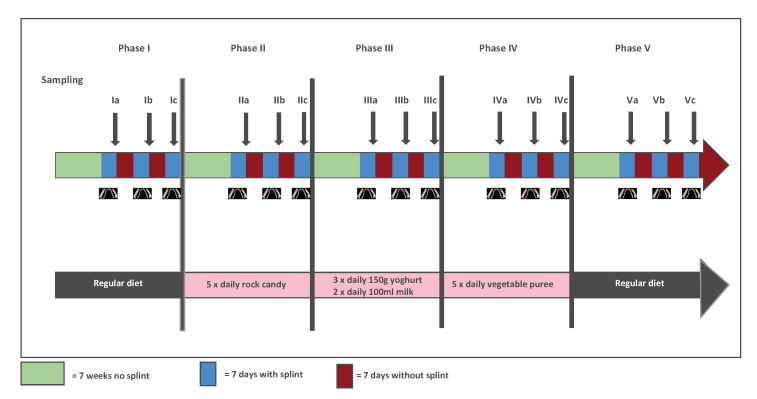
Study design: Description of the sampling and nutrition during the five phases.

**Figure 2 antibiotics-09-00479-f002:**
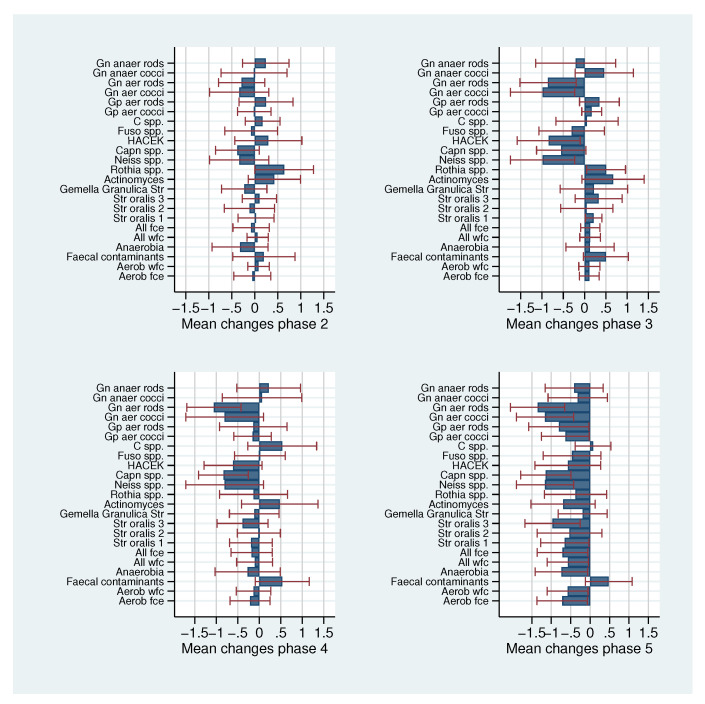
Mean changes of log bacteria concentrations in comparison to baseline with 95% CI.

**Figure 3 antibiotics-09-00479-f003:**
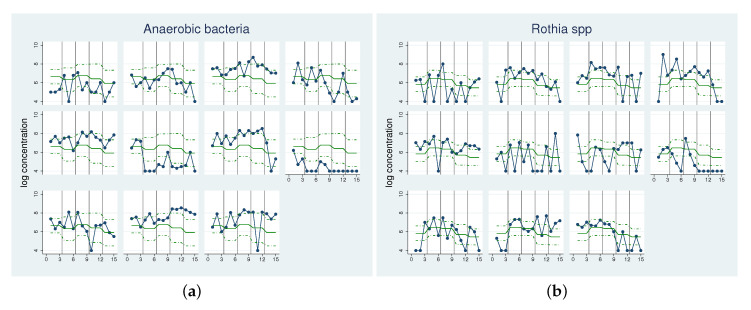
Example scatter plot of individual bacterial concentrations over time. Concentrations (in log-10 colony-forming unit (CFU) per mL) of anaerobic bacteria (**a**) and *Rothia* spp. (**b**) for each of the 11 participants over the five phases. Green line: mean value per phase, dotted green line: mean value ± sd per phase, grey line: end of the phase.

**Figure 4 antibiotics-09-00479-f004:**
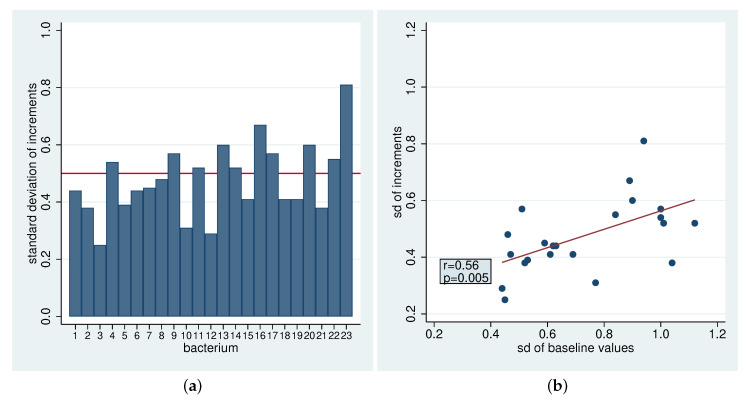
(**a**): Standard deviations of the increments (**b**): Standard deviations of the increments as a function of standard deviations of initial values with correlation coefficient r and corresponding *p*-value.

**Figure 5 antibiotics-09-00479-f005:**
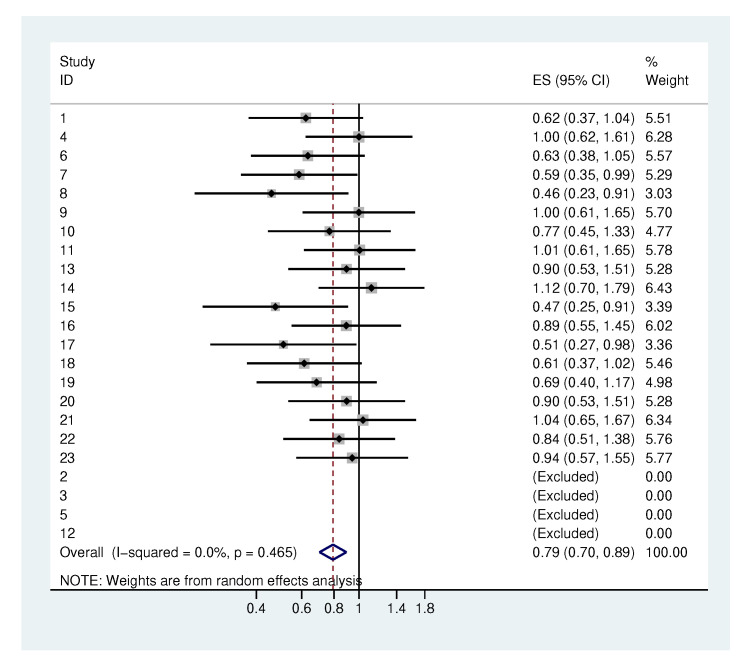
Meta-analysis of the standard deviations of the initial values. Each study represents one bacterial category.

**Figure 6 antibiotics-09-00479-f006:**
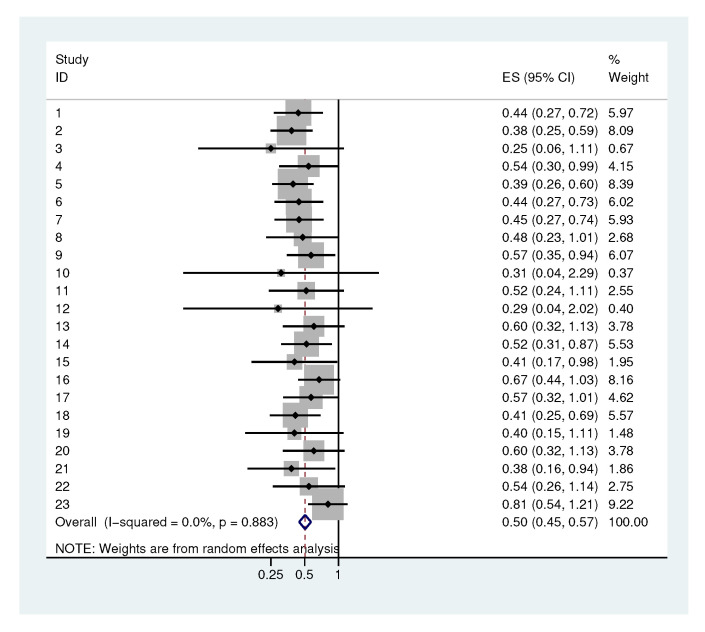
Meta-analysis of the standard deviations of the increments. Each study represents one bacterial category.

**Figure 7 antibiotics-09-00479-f007:**
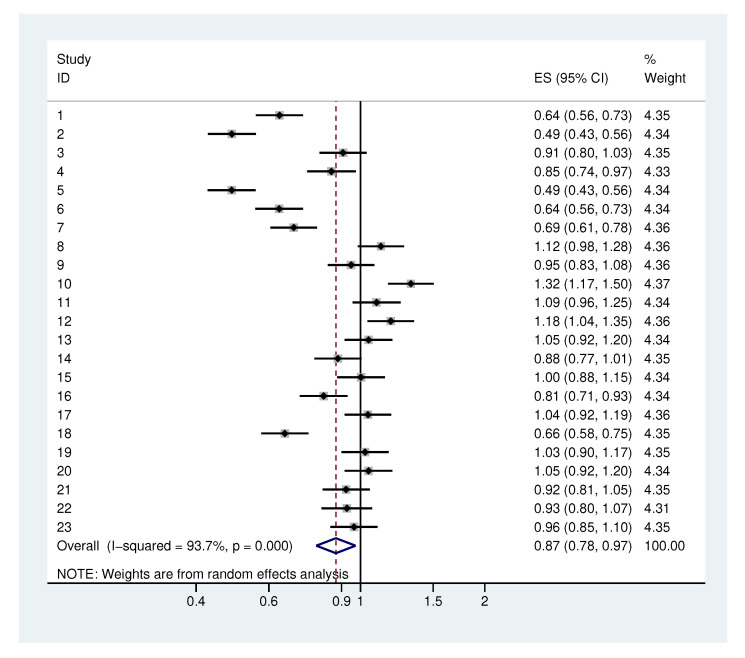
Meta-analysis of the standard deviations of the residuals. Each study represents one bacterial category.

**Figure 8 antibiotics-09-00479-f008:**
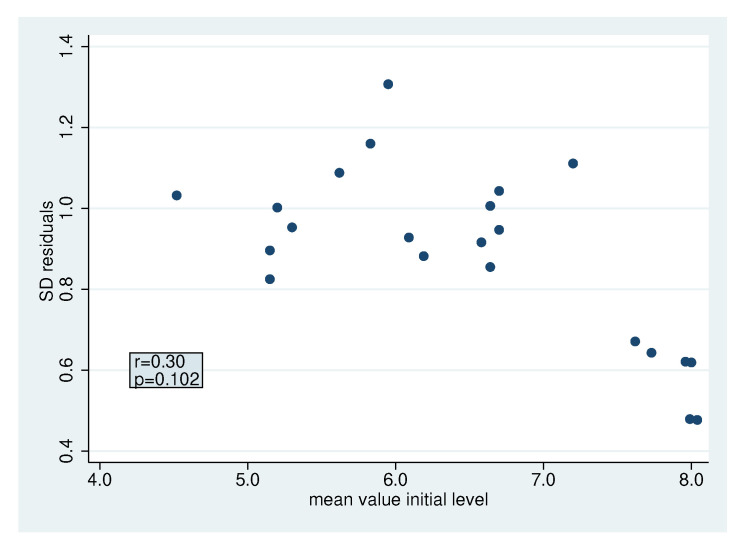
Relation of mean value of initial level and standard deviation of the residuals with correlation coefficient r and corresponding *p*-value.

**Table 1 antibiotics-09-00479-t001:** Model-based estimates for the example: Mean values and standard deviations of the initial values and increments for anaerobic bacteria and *Rothia* spp. with 95% CI. ${\hat{\sigma }}_{\Delta }$ indicates the common SD for the increments of phase 2–5. $\eta_{i}$ coherence of the individual changes with the mean change per phase.

Estimate	*Anaerobic bacteria*		*Rothia* spp.	
	**Estimate**	**95% CI**	**Estimate**	**95% CI**
Initial Values				
${\hat{\mu }}_{\alpha }$	6.644	[5.995, 7.294]	5.833	[5.355, 6.312]
${\hat{\sigma }}_{\alpha }$	1.000	[ 0.621, 1.611]	0.435	[0.212, 0.888]
Increments				
${\hat{\mu }}_{\Delta_{2}}$	−0.319	[−0.840, 0.201]	0.639	[0.042, 1.235]
${\hat{\mu }}_{\Delta_{3}}$	0.129	[−0.392, 0.649]	0.506	[−0.091, 1.102]
${\hat{\mu }}_{\Delta_{4}}$	−0.269	[−0.790, 0.251]	−0.132	[−0.729, 0.464]
${\hat{\mu }}_{\Delta_{5}}$	−0.741	[−1.262, −0.221]	−0.377	[−0.973, 0.219]
${\hat{\sigma }}_{\Delta_{2}}$	0.209	[0.004, 10.142]	0.000	[0, *∞* ]
${\hat{\sigma }}_{\Delta_{3}}$	0.000	[0.000, 0.404]	0.000	[0, *∞* ]
${\hat{\sigma }}_{\Delta_{4}}$	0.681	[0.321, 1.448]	0.605	[0, *∞* ]
${\hat{\sigma }}_{\Delta_{5}}$	0.720	[0.354, 1.463]	0.556	[0, *∞* ]
${\hat{\sigma }}_{\Delta }$	0.542	[0.297, 0.990]	0.290	[0.041, 2.018]
Coherence				
$\eta_{2}$	72.3		98.6	
$\eta_{3}$	59.5		96.1	
$\eta_{4}$	69.1		67.3	
$\eta_{5}$	91.5		90.5	

**Table 2 antibiotics-09-00479-t002:** Model-based estimates for all bacteria: Mean values and standard deviations of the initial values and increments. ${\hat{\sigma }}_{\Delta }$ indicates the common SD for the increments of phase 2–5. All abbreviations can be found in the list of abbreviations at the end of the manuscript.

Bacteria		Initial Values		Increments								
Nr	Name	${\hat{\mu }}_{\alpha }$	${\hat{\sigma }}_{\alpha }$	${\hat{\sigma }}_{\Delta }$	${\hat{\sigma }}_{\Delta_{2}}$	${\hat{\sigma }}_{\Delta_{3}}$	${\hat{\sigma }}_{\Delta_{4}}$	${\hat{\sigma }}_{\Delta_{5}}$	${\hat{\mu }}_{\Delta_{2}}$	${\hat{\mu }}_{\Delta_{3}}$	${\hat{\mu }}_{\Delta_{4}}$	${\hat{\mu }}_{\Delta_{5}}$
1	Aerob fce	7.96	0.62	0.44	0.35	0	0.43	0.81	−0.06	0.11	−0.21	−0.72
2	Aerob wfc	7.99	0.52	0.38	0	0	0.38	0.68	0.08	0.11	−0.13	−0.58
3	Faecal contaminants	5.15	0.45	0.25	0.58	0	0.28	0	0.20	0.50	0.54	0.48
4	Anaerobic bacteria	6.64	1.00	0.54	0.21	0	0.68	0.72	−0.32	0.13	−0.27	−0.74
5	All wfc	8.04	0.53	0.39	0	0	0.41	0.70	0.06	0.13	−0.11	−0.57
6	All fce	8.00	0.63	0.44	0.33	0	0.45	0.82	−0.08	0.13	−0.18	−0.71
7	*Str oralis* 1	7.62	0.59	0.45	0.33	0	0.56	0.79	0.03	0.21	−0.19	−0.65
8	*Str oralis* 2	7.20	0.46	0.48	0	0.55	0.30	0.80	−0.12	0.05	−0.01	−0.53
9	*Str oralis* 3	6.70	1.00	0.57	0.28	0.32	0.65	0.87	0.10	0.33	−0.39	−0.97
10	*Gem Granulicatella Str*	5.95	0.77	0.31	0	0.62	0.33	0.45	−0.23	0.22	−0.12	−0.19
11	*Act*	5.62	1.01	0.52	0	0.56	0.62	0.67	0.43	0.67	0.48	−0.70
12	*Rothia* spp.	5.83	0.44	0.29	0	0	0.60	0.56	0.64	0.51	−0.13	−0.38
13	*Neiss* spp.	6.70	0.90	0.60	0.48	0.56	0.82	0.56	−0.34	−0.99	−0.80	−1.16
14	*Capn* spp.	6.19	1.12	0.52	0.48	0.37	0.47	0.67	−0.38	−0.55	−0.83	−1.14
15	HACEK	5.20	0.47	0.41	0.59	0	0.45	0.42	0.30	−0.84	−0.61	−0.57
16	*Fuso* spp.	5.15	0.89	0.67	0	0.80	0.32	0.81	−0.08	−0.30	0.02	−0.47
17	*C* spp.	4.52	0.51	0.57	0	0.80	0.83	0.23	0.17	0.06	0.54	0.08
18	Gp aer cocci	7.73	0.61	0.41	0.26	0	0.40	0.77	−0.01	0.17	−0.15	−0.64
19	Gp aer rods	6.64	0.69	0.41	0	0	0.64	0.71	0.24	0.35	−0.14	−0.81
20	Gn aer cocci	6.70	0.90	0.60	0.48	0.56	0.82	0.56	−0.34	−0.99	−0.80	−1.16
21	Gn aer rods	6.58	1.04	0.38	0	0.30	0.32	0.67	−0.29	−0.86	−1.05	−1.35
22	Gn anaer cocci	6.09	0.84	0.55	0.35	0	0.81	0.55	−0.02	0.46	0.06	−0.32
23	Gn anaer rods	5.30	0.94	0.81	0	1.15	0.67	0.8 0	0.24	−0.21	0.22	−0.41

**Table 3 antibiotics-09-00479-t003:** Coherence of the individual values with the mean effect per phase. In addition the estimated residual standard deviation $\sigma_{res}$ is given. * indicates a significant effect at the 5% level from model-based estimates. Bacteria with large mean changes in Figure 2 are shaded gray. All abbreviations can be found in the list of abbreviations at the end of the manuscript.

Bacteria	$\eta_{2}$	$\eta_{3}$	$\eta_{4}$	$\eta_{5}$	$\sigma_{\mathrm{res}}$
Aerob fce	55.4	59.9	68.3	94.9 *	0.62
Aerob wfc	58.3	61.4	63.4	93.7 *	0.49
Faecal contaminants	78.8	97.7 *	98.5 *	97.3 *	0.91
Anaerobic bacteria	72.3	59.5	69.1	91.5 *	0.85
All wfc	56.1	63.1	61.1	92.8 *	0.49
All fce	57.2	61.6	65.9	94.7 *	0.64
*Str oralis* 1	52.7	68.0	66.4	92.6 *	0.69
*Str oralis* 2	59.9	54.1	50.8	86.5	1.12
*Str oralis* 3	57.0	71.9	75.3	95.6 *	0.95
*Gem Granulicatella* Str	77.1	76.1	65.1	73.0	1.32
*Act*	79.6	90.1 *	82.2	91.1 *	1.09
*Rothia* spp.	98.6 *	96.1	67.3	90.5	1.18
*Neiss* spp.	71.5	95.1 *	90.9 *	97.3 *	1.05
*Capn* spp.	76.8	85.8 *	94.5 *	98.6 *	0.88
HACEK	76.8	98.0 *	93.2 *	91.8 *	1.00
*Fuso* spp.	76.8	67.3	51.2	75.9	0.81
*C* spp.	61.7	54.2	82.8	55.6	1.04
Gp aer cocci	51.0	66.1	64.3	94.1 *	0.66
Gp aer rods	72.1	80.3	63.4	97.6 *	1.03
Gn aer cocci	71.5	95.1 *	90.9 *	97.3 *	1.05
Gn aer rods	77.7	98.8 *	99.7 *	100 *	0.92
Gn anaer cocci	51.5	79.9 *	54.3	72.0	0.93
Gn anaer rods	61.6	60.2	60.7	69.4	0.96

**Table 4 antibiotics-09-00479-t004:** Sample size and number of observations for $\sigma_{\Delta }=0.5$ and different μ, $\sigma_{res}$ and *R*.

*μ*	R	Sample Size /Number Observations	
		$\sigma_{\mathrm{res}}=0.6$	$\sigma_{\mathrm{res}}=1$
0.9	1	12/12	24/24
	2	9/18	15/30
	3	7/21	11/33
	4	7/28	10/40
0.7	1	18/18	39/39
	2	12/24	23/46
	3	10/30	17/51
	4	9/36	15/60
0.5	1	33/33	73/73
	2	22/44	42/84
	3	18/54	31/93
	4	16/64	26/104

**Table 5 antibiotics-09-00479-t005:** Coherence η for some combinations of μ (mean change) and σ (standard deviation of increments).

	σ		
μ	0.3	0.5	0.7
0.3	84.1	72.6	66.6
0.5	95.2	84.1	76.2
0.7	99.0	91.9	84.1
0.9	99.9	96.4	90.1

## Data Availability

The raw data, i.e., the concentrations of the individual bacteria per participant, phase and repetition in CFU/ml can be found as supplementary data.

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
