# Peer review of "Analysing the Relationship between Nutrition and the Microbial Composition of the Oral Biofilm—Insights from the Analysis of Individual Variability"

_antibiotics, 2020, doi:10.3390/antibiotics9080479_

Round 1

Reviewer 1 Report

I understood this study is a reanalysis of the published article of [ref.4], some new results were obtained. However, for readers without enough knowledge of [ref.4], it is very difficult to follow this study. Some concerns and comments to improve this work. 

1. More background information from [ref.4] is highly required. For instance, how all these bacteria were isolated, identified and classified.

2. In general, the structure of this manuscript requires improvement.

3. I suggest re-structing the abstract, too many technical details in the current version, which flooded key results and conclusions.

4. The main results of [ref.4] should be described in the introduction.

5. Since the data/samples were from another study [ref.4]. Then what is the major difference between this study and the published results? I suggest to present these important points in the abstract.

6. The same for introduction. many introductory contents from results section can be moved to introduction.

7. A direct repeat of this sentence “The influence of a change in nutrition on the oral microbiota has been discussed in the literature, but usually only changes of population mean values are reported.” In abstract and introduction, please reword them.

8. Why “the inter-individual discrepancies among patients” is important? How it is advanced over the standard mean value changes?

9. 11 adults times 5 phases gives only 55 sampling points. Please convince readers the sample size is reasonable and scientifically sound to be used for modeling and further analysis.

10. What does "to a more targeted planning of new studies." in the conclusion mean?

11. "The data have not been completely analyzed until now. Further publications are in preparation." This statement must be changed. I insist that authors provide data that they used in this study.

Author Response

Dear Reviewer 1,

thank you very much for the many helpful comments.

Review 1:

I understood this study is a reanalysis of the published article of [ref.4], some new results were obtained. However, for readers without enough knowledge of [ref.4], it is very difficult to follow this study. Some concerns and comments to improve this work. 

Response: Thank you for this comment, we try to improve the readability. It is true that [ref.4] refers to the same patient collective, but only to two phases and to an evaluation using the sequencing technique. This article deals with different data from the same study using another evaluation method, namely the culture technique, which describes all five diet phases. We clarified this in the manuscript and wrote (L82-86):

“The previous study [ref.4] refers to the same patient collective, but used a different technique (sequencing) to measure the bacterial composition, thereby including only two diet phases in the analysis. The study showed that in the sucrose phase (phase 2) the microbial community composition was significantly different than in phase 1 (baseline). Especially the abundance of oral streptococci was significantly increased.”

  1. More background information from [ref.4] is highly required. For instance, how all these bacteria were isolated, identified and classified.

Response 1: The focus of this manuscript was the statistical evaluation of the data. Nevertheless, we provided a description of the method by adding the following text to the method section (L87-98): The bacteria were isolated and identified as previously described in detail (Schirrmeister et al., 2009, Bernardi et al., 2020). The vials containing the samples were thawed at 36°C in a water bath and vortexed for 30–45 s. For the isolation and identification of the microorganisms, 100 µl of the undiluted sample and serial dilutions thereof were plated on different agar plates. The serial dilutions (10−1 to 10−7) were prepared in peptone yeast medium (PY). Each dilution was plated on Columbia blood agar plates (CBA) at 37°C and 5%–10% CO2 atmosphere for 5 days to cultivate aerobic species and on yeast-cysteine blood agar plates (HCB) to cultivate anaerobic bacteria at 37°C for 10 days. The grown bacterial colonies were phenotypically evaluated and counted. Hence, the number of colony forming units (CFUs) per ml in the original sample was calculated. Pure cultures of all colony types were sub-cultivated and analyzed by MALDI-TOF (MALDI Biotyper, Bruker Daltonik GmbH, Bremen, Germany), as described earlier in detail (Anderson et al., 2014).

References:

Bernardi, S., Karygianni, L., Filippi, A., Anderson, A.C., Zürcher, A., Hellwig, E., Vach, K., Macchiarelli, G., Al-Ahmad, A. Combining culture and culture-independent methods reveals new microbial composition of halitosis patients' tongue biofilm. Microbiologyopen. 2020,9(2),e958

Anderson, A.C., Sanunu, M., Schneider, C., Clad, A.,Karygianni, L., Hellwig, E. and Al-Ahmad, A. Rapid species-level identification of vaginal and oral lactobacilli using MALDI-TOF MS analysis and 16S rDNA sequencing. BMC Microbiol. 2014,14:312.    

 Schirrmeister,J.F., Liebenow, A.L., Pelz, K., Wittmer, A., Serr, A., Hellwig, E. and Al-Ahmad, A.   New bacterial compositions in root-filled teeth with periradicular lesions. J Endod. 2009, 3,169–174.  

  1. In general, the structure of this manuscript requires improvement.

Response 2: Thank you for this comment, we changed the abstract, introduction and part of the results and improved the readability.

  1. I suggest re-structing the abstract, too many technical details in the current version, which flooded key results and conclusions.

Response 3: Thank you for this comment, we deleted some technical details and mentioned more results. The abstract was changed to:

The influence of a change in nutrition on the oral microbiota has been discussed in literature,  but usually only changes of population mean values are reported. This paper introduces simple methods to also analyse and report the variability of patients’ reactions considering data from the culture analysis of oral biofilm. The framework was illustrated by an experimental study exposing eleven participants to different nutrition schemes in 5 consecutive phases. Substantial inter-individual variations in the individual reactions were observed. A new coherence index made it possible to identify 14 instances where the direction of individual changes tended to coincide with the direction of the mean change with more than 95% probability. In these instances it can be expected that the vast majority of the general population will experience a change in the concentration as indicated by the change in mean values if exposed to a corresponding diet. The heterogeneity in variability across different bacteria species was limited. This allowed us to develop recommendations for sample sizes in future studies. For studies measuring the concentration change of bacteria as a reaction to nutrition change, the use of replications and analysis of the variability is recommended. In order to detect moderate effects of a changed nutrition on the concentration of single bacterial taxa 30 participants with three repetitions are often adequate.

  1. The main results of [ref.4] should be described in the introduction.

Response 4: Thank you for this comment, we added (L82-86):

 “The previous study [ref.4] refers to the same patient collective, but used a different technique (sequencing) to measure the bacterial composition, thereby including only two diet phases in the analysis. The study showed that in the sucrose phase (phase 2) the microbial community composition was significantly different than phase 1(baseline).  Especially the abundance of oral streptococci was significantly increased.”

  1. Since the data/samples were from another study [ref.4]. Then what is the major difference between this study and the published results? I suggest to present these important points in the abstract.

Response 5: The previous study [ref. 4] referred to a completely different method and dealt only with diet phases 1 and 2. The main results have now been added to the introduction (see point 4).

  1. The same for introduction. many introductory contents from results section can be moved to introduction.

Response 6: Thank you for this proposal. We have rewritten the introduction.

  1. A direct repeat of this sentence “The influence of a change in nutrition on the oral microbiota has been discussed in the literature, but usually only changes of population mean values are reported.” In abstract and introduction, please reword them.

Response 7: Thank you for this point, we changed it to (L21):

“In recent years, the influence of nutrition on oral microbiota has often been discussed in  literature.“

  1. Why “the inter-individual discrepancies among patients” is important? How it is advanced over the standard mean value changes?

Response 8:  When reporting only a mean value of changes, there is some danger that the value is implicitly interpreted as value which applies to all patients, or that at least the sign of the change (increase or decrease) applies to all or the vast majority of patients. However, this can, but does not need to be true. By considering the inter-individual variation among patients, we can depict to some degree, to which extent such an assumption is justified. The proposed h-values seem to be one simple way to achieve this. In this respect, the inter-individual differences allow for the correct interpretation of the results.

We hope by presented analyses based on the mean values as well as based on the h values, the additional information provided by considering individual variation becomes more obvious.  

  1. 11 adults times 5 phases gives only 55 sampling points. Please convince readers the sample size is reasonable and scientifically sound to be used for modeling and further analysis.

Response 9:  We agree, that this is an important point. In our paper we have a sample size calculation (Table 4). Here it can be seen, that a sample size of 11 participants with 3 repetitions is large enough to detect a mean difference of 0.9 if the variation is large (sd of residuals =1) or a moderate mean difference of 0.5 if the variation is not too large (sd of residuals =0.6). This was already stated in the manuscript.

  1. What does "to a more targeted planning of new studies." in the conclusion mean?

Response 10: The presented insights into the relations between the overall sample size, the number of replicates and the power to demonstrate should influence the planning of new studies in the sense of an optimal division of available resources on number of study participants and number of replicates. Typically, it is easier and cheaper to increase the number of replicates than the number of probands. The formulas provided enable balancing the costs against the gain in power and to find an optimal solution. In this sense we can adapt the planning of a study to the target to maximize statistical power for a fixed budget.

  1. "The data have not been completely analyzed until now. Further publications are in preparation." This statement must be changed. I insist that authors provide data that they used in this study.

Response 11: We have made the data available as supplementary file data_article1.csv.

Reviewer 2 Report

The methods of the paper need to be described more in details.

The authors mentioned yoghurt and milk were used as the nutrition. What type of yoghurt and milk were used? Different type/brand may have significant difference in composition. Please provide details.

What type of splint was used? Rigid or flex or semi-flex splint? Were the splint worn all the time during the 7 days or they were removed at night?

Please provide the standard deviation values in Figures 2 and 3.

Please discuss the limitation of wearing in-situ splint in the present study.

Please discuss how can the nutrition influence the oral biofilm from different origins such as cariology, periodontal, and endodontic sources.

Author Response

Dear reviewer,

thank you very much for the many helpful comments.

The methods of the paper need to be described more in details.

The authors mentioned yoghurt and milk were used as the nutrition. What type of yoghurt and milk were used? Different type/brand may have significant difference in composition. Please provide details.

Response: Thank you for this hint. We added in the M&M section (L56,57):

The yoghurt and milk that were used contained both 1.5% fat and were purchased from Schwarzwaldmilch GmbH, Freiburg, Germany

What type of splint was used? Rigid or flex or semi-flex splint? Were the splint worn all the time during the 7 days or they were removed at night?

Response: Thank you for this comment. Individual upper jaw rigid acrylic appliances were fabricated for each study participant (Ref. 4). The splint system was worn at all times except during meals and dental hygiene. This information has been included in the manuscript L65-67.

Please provide the standard deviation values in Figures 2 and 3.

Response: We added the standard deviations in both Figures.

Please discuss the limitation of wearing in-situ splint in the present study.

Response: It has been shown in many previous own studies that supragingival oral biofilm was cultivated sufficiently on enamel slabs, which are fixed in splint systems. However, splints still remain a model system which simulates the natural conditions of biofilm formation in the oral cavity. It cannot be excluded that the salivary pellicle and the subsequent formation on bovine enamel slabs could be slightly altered. Additionally, during meals the splint systems had to be taken out and stored in saline solution (0.9%). This information has been included in the discussion (L360-366).

Please discuss how can the nutrition influence the oral biofilm from different origins such as cariology, periodontal, and endodontic sources.

Response: Thank you for this suggestion. We added in the discussion (L335-340):

Nutrition can influence the oral biofilm composition and thus, the onset of oral diseases. Specifically, the content of fermentable carbohydrates is crucial in the process of cariogenic demineralisation.  (Nyvad, B., Takahashi, N. Integrated hypothesis of dental caries and periodontal diseases. J Oral Microbiol. 2020 Jan 7;12(1):1710953).

However, recent studies also showed an influence of sugar on gingival inflammation which might be etiologically related to both local and systemic effects like elevated blood sugar (Hujoel, P. Dietary carbohydrates and dental-systemic diseases. J Dent Res. 2009 Jun;88(6):490-502. / Woelber,J.P., Tennert, C. Chapter 13: Diet and Periodontal Diseases. Monogr Oral Sci. 2020;28:125-133).

Secondly, nutrition can have antibacterial and biofilm-inhibiting properties, which can be attributed to polyphenols  (Wittpahl, G., Kölling-Speer, I., Basche, S., Herrmann, E., Hannig, M., Speer, K., Hannig, C. The Polyphenolic Composition of Cistus incanus Herbal Tea and Its Antibacterial and Anti-adherent Activity against Streptococcus mutans. Planta Med. 2015 Dec;81(18):1727-35)

Furthermore, nitrates can have both anti-cariogenic and anti-inflammtory effects on gingivitis (Scoffield, J., Michalek, S., Harber, G., Eipers, P., Morrow, C., Wu, H. Dietary Nitrite Drives Disease Outcomes in Oral Polymicrobial Infections. J Dent Res. 2019;98(9):1020-1026.    / Jockel-Schneider, Y., Goßner, S.K., Petersen, N., Stölzel, P., Hägele, F., Schweiggert, R.M., Haubitz, I., Eigenthaler, M., Carle, R., Schlagenhauf, U. Stimulation of the nitrate-nitrite-NO-metabolism by repeated lettuce juice consumption decreases gingival inflammation in periodontal recall patients: a randomized, double-blinded, placebo-controlled clinical trial. J Clin Periodontol. 2016 Jul;43(7):603-8.)

Reviewer 3 Report

The overarching issue of this manuscript is the lack of clear description in the different sections, especially the microbiology parts, although the modelling and the variability parts are well supported with details. 

An abstract must be a concise synopsis of the key parts of the research, at least contains a brief results summary supported by data. There are no study results mentioned in this section supported by evidence.

L28: please explain what is a “motivating example”.

L35-51: this section needs to be completely rewritten. 

Some details: The funding agency goes in the dedicated paragraph. Explain the following: “lead-in-phase” (is it different from “regular diet” in the figure1?; “high-fiber diet”; “in-situ splint system”.

L52: provide a reference formatting according to the standards of the journal.

L56-57: The methods of detection of the bacteria must be clear and presented at least in a table with the standards used/protocol and control used by the authors.

L93-101: the section needs to be rewritten. What is the rationale for this “Standard deviations are difficult to interpret for non-statisticians.”? The manuscript is presented to a specialised community, not a generic audience.

L136: format the reference.

L144: The choice of clustering some of the microorganisms must be previously described, including the referring in the introduction and material and methods sections.

Table 5 should be inserted into the main text close to their first citation.

Author Response

Dear Reviewer,

thank you very much for the many helpful comments.

The overarching issue of this manuscript is the lack of clear description in the different sections, especially the microbiology parts, although the modelling and the variability parts are well supported with details. 

 An abstract must be a concise synopsis of the key parts of the research, at least contains a brief results summary supported by data. There are no study results mentioned in this section supported by evidence.

Response:  Thank you for this comment. We rearranged the abstract and introduction.

L28: please explain what is a “motivating example”.

Response:  The intention of the paper was to use example data to explain how important it is not only to calculate mean values and standard deviations, but also to consider other measures. We removed the word “motivating” now.

L35-51: this section needs to be completely rewritten. 

Some details: The funding agency goes in the dedicated paragraph. Explain the following: “lead-in-phase” (is it different from “regular diet” in the figure1?; “high-fiber diet”; “in-situ splint system”.

Response: The funding was now only mentioned in the dedicated paragraph and the paragraph was rewritten and changed to (L50-67):

In this study with eleven adults (5 male, 6 female) aged between 21 and 56 years, data concerning nutrition and the microbiota in their oral biofilm were collected over 15 months [4]. The participants ran through five phases, each of which lasted 3 months, following a specific nutritional protocol. In a first lead-in-phase, the participants kept their regular diet, which served as baseline. ….. An in-situ splint system was used to sample the dental plaque. Individual upper jaw rigid acrylic   appliances were fabricated for each study participant [4]. The splint system was worn at all times except during meals and dental hygiene.

L52: provide a reference formatting according to the standards of the journal.

Response:  We added the references:

Bernardi, S., Karygianni, L., Filippi, A., Anderson, A.C., Zürcher, A., Hellwig, E., Vach, K., Macchiarelli, G., Al-Ahmad, A. Combining culture and culture-independent methods reveals new microbial composition of halitosis patients' tongue biofilm. Microbiologyopen. 2020,9(2),e958

Anderson, A.C., Sanunu, M., Schneider, C., Clad, A.,Karygianni, L., Hellwig, E. and Al-Ahmad, A. Rapid species-level identification of vaginal and oral lactobacilli using MALDI-TOF MS analysis and 16S rDNA sequencing. BMC Microbiol. 2014,14:312.    

 Schirrmeister,J.F., Liebenow, A.L., Pelz, K., Wittmer, A., Serr, A., Hellwig, E. and Al-Ahmad, A.   New bacterial compositions in root-filled teeth with periradicular lesions. J Endod. 2009, 3,169–174. 

L56-57: The methods of detection of the bacteria must be clear and presented at least in a table with the standards used/protocol and control used by the authors.

Response:  A description of the method used for bacterial detection has now been added to the method section (see our response to Reviewer 1) (L77-87): The bacteria were isolated and identified as previously described in detail (Schirrmeister et al., 2009, Bernardi et al., 2020). The vials containing the samples were thawed at 36°C in a water bath and vortexed for 30–45 s. For the isolation and identification of the microorganisms, 100 µl of the undiluted sample and serial dilutions thereof were plated on different agar plates. The serial dilutions (10−1 to 10−7) were prepared in peptone yeast medium (PY). Each dilution was plated on Columbia blood agar plates (CBA) at 37°C and 5%–10% CO2 atmosphere for 5 days to cultivate aerobic species and on yeast-cysteine blood agar plates (HCB) to cultivate anaerobic bacteria at 37°C for 10 days. The grown bacterial colonies were phenotypically evaluated and counted. Hence, the number of colony forming units (CFUs) per ml in the original sample was calculated. Pure cultures of all colony types were sub-cultivated and analyzed by MALDI-TOF (MALDI Biotyper, Bruker Daltonik GmbH, Bremen, Germany), as described earlier in detail (Anderson et al., 2014).

L93-101: the section needs to be rewritten. What is the rationale for this “Standard deviations are difficult to interpret for non-statisticians.”? The manuscript is presented to a specialised community, not a generic audience.

Response:  In my more than 20 years of working with medical doctors I have often experienced that the interpretation of population standard deviations can be challenging, as – in contrast to p-values – there is no simple rule for good or poor values. Moreover, population standard deviations are often confused with standard deviations of estimated mean values, i.e. standard errors. Based on this experience, this section tries to offer some assistance to those readers who are not familiar with the interpretation of population standard deviations. But of course there are also readers who are familiar with these terminologies, but who can certainly ignore this section. We tried to take this into account by adding the phrase “are OFTEN difficult”.

L136: format the reference.

Response:  The reference was changed accordingly.

L144: The choice of clustering some of the microorganisms must be previously described, including the referring in the introduction and material and methods sections.

Response:  We have rewritten the introduction and material and methods sections.

Table 5 should be inserted into the main text close to their first citation.

Response:  Thank you for this hint. We tried to implement this.

Reviewer 4 Report

The manuscript titled, "Analysing the relationship between nutrition and the
microbial composition of the oral biofilm - insights from the analysis of individual variability" by Hellwig et al, is an interesting concept and in-depth study on the correlations between individual nutrition change to bacteria concentration.

The results are well documented and analysed.

This study will definitely be helpful for the development of antibacterial compounds targeted against specific oral bacteria or biofilm.

Author Response

Dear reviewer,

thank you very much for the kind words.

Reviewer 5 Report

[Suggestions]
The authors need to clarify the reason why the manuscript was submitted to the "Antibiotics". Were the contents of the manuscript related to some antibiotics?

The referee is interested in the Reference #4 by the authors. In the Introduction, the authors need to clarify the differences (from the Ref #4) and the rationale of the present study in further detail.

Author Response

Dear reviewer,

thank you very much for the many helpful comments.

The authors need to clarify the reason why the manuscript was submitted to the "Antibiotics". Were the contents of the manuscript related to some antibiotics?

Response: In this manuscript we presented an in-depth study on the correlations between individual nutrition change and bacterial concentration. Keeping in mind that pathological shifts in biofilm composition could trigger the onset of oral diseases, one could assume that this study gives new statistical insights in the relationship between nutrition and the microbial composition. This can be helpful for the development of diet habits that promote the establishment of a healthy microbial flora and can therefore prevent the initiation of oral diseases such as caries and periodontitis. The report can also be indirectly helpful for the development of antibacterial compounds and alternative treatment methods such as photodynamic antimicrobial therapy (PDT) targeted against specific oral bacteria or oral biofilm. Alternative biofilm treatment methods such as PDT or natural compounds may alter the oral biofilm composition as well. Therefore the special issue “Oral Microorganisms and Inactivation of Oral Biofilms“ of Antibiotics was chosen.

The referee is interested in the Reference #4 by the authors. In the Introduction, the authors need to clarify the differences (from the Ref #4) and the rationale of the present study in further detail.

Response: [ref.4] refers to the same patient collective, but only to two phases and to an evaluation using the sequence technique. This article deals with different data from the same study using culture technique and all five diet phases are examined. We clarified this in the manuscript and added (72-75):

“The previous study [ref.4] refers to the same patient collective, but used a different technique (sequencing) to measure the bacterial composition, thereby including only two diet phases in the analysis. The study showed that in the sucrose phase (phase 2) the microbial community composition was significantly different than in phase 1 (baseline). Especially the abundance of oral streptococci was significantly increased.”

Round 2

Reviewer 1 Report

Most of my major concerns have been addressed. I don't have more other comments.

Reviewer 3 Report

Please state in the introduction and abstract how the study meets the scope of the journal/special issue “Oral Microorganisms and Inactivation of Oral Biofilms“. I am aware that the results presented can be helpful for the "development of dietary habits that promote the establishment of healthy microbial flora", but this report must be stated more clearly in order to better understand the importance of this study in the area of "antibacterial compounds and alternative treatment methods such as photodynamic antimicrobial therapy (PDT) targeted against specific oral bacteria or oral biofilm."

A few corrections below:

Check reference format, in particular, the numbering system.

Line 27: “The more uniform a change” capital letter not required

Line 87: check the journal’s suggestion for formulae formats: 4.72 ∗ 109

Figure 4 and 8: report the number significant

L378 specify which data can be found as supplement material.

Author Response

Review 3 (Round 2):

Please state in the introduction and abstract how the study meets the scope of the journal/special issue “Oral Microorganisms and Inactivation of Oral Biofilms“. I am aware that the results presented can be helpful for the "development of dietary habits that promote the establishment of healthy microbial flora", but this report must be stated more clearly in order to better understand the importance of this study in the area of "antibacterial compounds and alternative treatment methods such as photodynamic antimicrobial therapy (PDT) targeted against specific oral bacteria or oral biofilm."

Response: Thank you for this comment we changed abstract and introduction.

The following text has now been added to the abstract (L13-16): “Insights in the relationship between nutrition and the microbial composition can be helpful for the development of dietary habits that promote the establishment of a healthy microbial flora and can therefore prevent the initiation of oral diseases such as caries and periodontitis.”

The following text has now been added to the introduction (L52-59):

“Keeping in mind that shifts in biofilm composition towards a dysbiotic microbiota could trigger the onset of oral diseases, one could assume that this study gives new statistical insights in the relationship between nutrition and the microbial composition. This can be helpful for the development of dietary habits that promote the establishment of a healthy microbial flora and can therefore prevent the initiation of oral diseases such as caries and periodontitis. The report can also be indirectly helpful for the development of antibacterial compounds and alternative treatment methods such as antimicrobial photodynamic therapy (aPDT) targeted against specific oral bacteria or oral biofilm. Alternative biofilm treatment methods such as aPDT or natural compounds may alter the oral biofilm composition as well.”

A few corrections below:

Check reference format, in particular, the numbering system.

Response: Thank you for this hint. We checked the references, especially the numbers of the volumes are now italic.

Line 27: “The more uniform a change” capital letter not required

Response: We changed the sentence to: “ … the more uniform …”.

Line 87: check the journal’s suggestion for formulae formats: 4.72 ∗ 109

Response: The formula was changed accordingly (L 99).

Figure 4 and 8: report the number significant

Response: The p-values were added in Figure 4 and 8.

L378 specify which data can be found as supplement material.

Response: We specified the data and changed this paragraph to (L 390-391):

“The raw data, i.e. the concentrations of the individual bacteria per participant, phase and repetition in CFU/ml can be found as supplementary data.“

Reviewer 5 Report

I understand the responses by the authors.

Author Response

Thank you very much.

Round 3

Reviewer 3 Report

The authors accordingly motivated and updated the manuscript with a few corrections as suggested.